# Collaborative Ryukyuan Language Documentation and Reclamation

Madoka Hammine [1,*] and Martha Tsutsui Billins [2,*]

1    Faculty of International Studies, Meio University, Nago 905-8585, Japan
2    Department of Linguistics, California State University, Fresno, CA 93740, USA
*    Correspondence: m.hammine@meio-u.ac.jp (M.H.); marthatsutsuibillins@mail.fresnostate.edu (M.T.B.)

**Abstract:** Traditional "endangered" approaches in linguistics tend to impose Western epistemologies of languages on marginalized Indigenous language communities such as the Ryukyus. Instead, by using a collaborative approach, we ask for a change of approach from research *on* the Ryukyus to research *with/for* the Ryukyus. This article is a reflective study of collaboration in particular cases. We aim to address the issues of relationality between communities and researchers—how can communities initiate work with like-minded linguists to suit their own needs? Thus, we respond to this question to open a conversation on why insider/outsider collaboration is essential. Using our experiences of carrying out our research in different parts of the Ryukyus reflectively, we aim to provide a practical guide for collaboration that is necessary for both the good of communities and the field of linguistics. Through continuous cooperation and collaboration, we can engage in active decolonization of the field of linguistics and language documentation. We suggest that decolonization cannot be achieved without collaborative and ethical research practices based on Indigenous epistemologies. We conclude the paper with ideas of research approaches based on Ryukyuan Indigenous epistemologies, which require a transformation from individual approaches to community-based-relational approaches.

**Keywords:** language documentation; language reclamation; Indigenous Research Methodologies; relationalities; equity; decolonization

## 1. Background

Discourses about "endangerment" and "language revitalization" have emerged since the beginning of the 1990s, offering a contrastive interpretation of history. After centuries of marginalization, assimilation and oppression, the "Indigenous languages" have been given special attention worldwide, and the paradigm of "endangerment" became the dominant and crucial call-to-action for linguists, scholars, nation states, and international organizations (see e.g., Boitel, forthcoming). The paradigm of language endangerment was to provide "solutions" to the problems of language loss through theoretical frameworks and financial support to research projects aimed at documenting and revitalizing Indigenous languages all over the globe. Along with this development, many studies have demonstrated that discourses of endangerment, far from having the emancipatory effects usually claimed, can contribute to the perpetuation of domination, discrimination, racial prejudice, and exploitation of Indigenous communities (see Costa 2016; Davis 2017; Duchêne and Heller 2008; Hill 2002; Muehlmann 2013; Roche 2020). The "endangerment" approach often attaches great importance to linguistic diversity, claiming these languages as a global treasure of *all* humanity. Scholars also criticize the "endangerment" approach for tending to focus on documenting speech of native speakers of minoritized languages by enumeration, that is, counting numbers of such languages (e.g., Hill 2002), as well as linguistic extraction, treating participants solely as data providers removed from their embodied lived experiences with their languages (e.g., Davis 2017). Behind these critics also lies the realm

of West-centric epistemologies to look at and understand languages. These discourses focus on defending minority "endangered" languages against the incursion of larger ones or on defending multilingualism based on a West-centric understanding of "languages" (Pennycook and Makoni 2019). Despite these critiques convincingly and repeatedly showing that thinking in terms of the "endangerment" approach is problematic and could have negative effects on Indigenous communities, this framework remains hegemonic among scholars and language activists. Instead, language documentation, reclamation and revitalization of Indigenous languages should be seen as the emancipation of minorities and their cultures on their own terms rather than on the terms of humanity (Hill 2002).

This situation proves true in the context of the Ryukyus. Since recognition by UNESCO in 2009, the discourse of language endangerment seems to have raised peoples' awareness about language "endangerment" (e.g., Okinawa Times 2014, 2015). Although people have become aware of "endangerment", language revitalization in many communities in the Ryukyus does not seem to emancipate language communities, speakers and learners. As researchers who have been working with different communities and islands of the Ryukyus, we show there is a need for changing research approaches, based on Indigenous decolonial research approaches. We further argue against the "endangerment" approach and support a collaborative relational approach in language documentation. We hope that this paper will provide two examples (our own) of how research in the Ryukyus can be conducted taking on a more collaborative approach based on Indigenous epistemologies and Indigenous methodologies. Thus, this paper relates to endangered language studies and, more specifically, Ryukyuan language endangerment studies, by presenting alternative practices in research in Ryukyuan language communities.

The reason for such a change traces back to a long multilayered history of colonization in the Ryukyus (McCormack and Norimatsu 2012). A symptom of this is that some Ryukyuan people themselves do not recognize their languages as distinct from Japanese. Rather, many Ryukyuans refer to their languages as *hōgen* ("dialect"), which likely comes from the dominant Japanese perspective based on a prevailing ideology that Ryukyuans' local languages are mere dialects of Japanese (Heinrich 2012; Lee 2009; Yasuda 1999). Traditionally, most work in the Ryukyus has centered on a single outsider linguist (i.e., someone not from the community in question), who plans a project alone with no or little input from the community and then goes into the community and collects data via extractive methods. Indeed, most language documentation has occurred this way, even beyond the Ryukyus, but there has not yet been a call for change, particularly in Ryukyu Studies, for more community-centered research approaches. Endangered languages can only be revitalized by communities, so it is essential that researchers and communities work together in the Ryukyus, where all varieties are endangered.

In this paper and our broader work, we focus on collaborative *relational* research, identifying some issues related to research ethics and principles we face during our fieldwork. While the traditional "endangerment" approach tends to be based on a colonial mindset, recent scholarship reveals that community-centeredness is increasingly being considered invaluable to language documentation in the field (Cameron et al. 1992; Dwyer 2010; Hermes and Engman 2017; Leonard 2018a, 2018b; Penfield et al. 2008; Yamada 2007). However, this shift has not yet fully taken place in the Ryukyus. As Cruz and Woodbury (2014) point out, it is very tricky to generalize. For instance, Cruz and Woodbury (2014) point out:

> There are differences among researchers themselves, their research agendas and goals, and their relationships to the community; differences among speakers and other local stakeholders and their ideas and visions for their linguistic heritages; and differences in the communicative roles or niches that languages may fill for their speakers. And any of these differences may respond to larger patterns and conditions holding in the region, nation-state, or even continent in which a language is spoken. (Cruz and Woodbury 2014, p. 1)

Therefore, we put forth our own particular case as a "detailed, reflective study of collaboration" (Cruz and Woodbury 2014, p. 1) to support our hypothesis that community-centeredness is invaluable in the Ryukyus as well. Taking on the role of "language activists" (in the sense of Florey et al. 2009), where we put forth "energetic action toward [preserving and promoting linguistic diversity/supporting language rights]" (p. 4), we discuss our experiences in Indigenous language communities and raise questions regarding what it means to be community-centered.

We write this article as two voices and two people to share stories of our research and reflect from these points of view on larger issues related to language research in the Ryukyus. Martha started her doctoral project of documenting the Setouchi Amami language as a PhD student at SOAS, University of London and continues to do so as a lecturer at the University of California, Fresno. Madoka started her doctoral project of Indigenous language revitalization in Finland focusing initially with Sámi language communities and then with the Miyara Yaeyama language communities in the Ryukyus. Acknowledging the challenges of conducting Ryukyuan language documentation and revitalization research with and for communities, we share our experiences as researchers and our processes of engagement and language documentation chronologically. This paper's aim is to open a dialogue between and amongst researchers and communities, rather than adhere to a traditional research design with clear-cut data. Accordingly, following this introduction, our presentation will be dialogic and discursive. We refer to ourselves asMadoka and Martha to discuss our experiences.

## 2. Our Collaboration Stories

We consider collaboration to be a key activity to decolonization in the Ryukyus, while understanding researcher positionalities is a prerequisite first step. Our work together began in 2018 through a course held at the University of Helsinki. The course focused on Ryukyuan languages. At that time, Martha was a PhD candidate at SOAS, University of London while Madoka was writing a doctoral dissertation at the University of Lapland in Finland. We met in a class led by Dr. Patrick Heinrich on Ryukyuan languages. While our projects focus on different communities in the Ryukyus, we started interacting with each other and sharing our thoughts on how researchers should work with communities.

Previously, scholars have discussed decolonization and emancipation of Ryukyuan Indigenous communities from viewpoints of local food practices (Chibana 2020), Okinawan literature (Young 2020), and feminist and demilitarization movements in Okinawa (Ginoza 2015, 2016; Sakuma 2021). Chibana (2020), for instance, argued that Ryukyuan islanders are not homogenizing Okinawan Indigeneity through everyday food practices, but rather, islanders are "creating and envisioning more sustainable and inclusive space while anchoring firmly within the ancestral land" (p. 3). Sakuma (2021) analyzed possibilities of "souvenirs of solidarity" in demilitarization movements on Okinawa Island based on shared experiences of protests (p. 11). These studies demonstrate social political power dynamics in the Ryukyus and how Indigenous communities experience power relationships in relation to Japan and the U.S. Similarly, these power dynamics are also present in linguistics.

In Ryukyuan linguistics and language research, there are a dearth of publications articulating how these social and political power dynamics affect researchers and how they conduct research in Indigenous language communities. Ishihara (2011) highlighted this in a comparison of research ethics of the Global North, comparing countries such as the U.S. and Canada to Japan (p. 77). In Japan, although the fields of medical studies and psychology consider research ethics to be important, many fields of humanities including linguistics traditionally did not have concrete guidelines or approaches regarding research ethics while in North America and Australia, researchers who work with endangered language communities usually have been required to pass examinations of research ethics from research ethics commissions in their home institutions. In that sense, Ishihara (2011)

asserts that researchers who work on Ryukyuan languages also need to consider a high standard of research ethics.

Based on Ishihara (2011), as one prerequisite to research ethics, we explore researcher positionalities in Ryukyuan linguistics. By understanding dominant, colonial, Japanese, and/or Western positions, researchers themselves can understand a possibility of a more collaborative way of approaching endangered language communities. Unless outside researchers work with "sensitivity and sympathy" (Fishman 1968, p. 6) toward language communities, I (Madoka) find such researchers quite difficult to work with because their lack of awareness and their insensitivity may alienate marginalized communities instead of including them equitably. Likewise, other Ryukyuans may have similar experiences with outsider researchers, which not only alienates community members but also limits the data quality that such projects have access to. Since our data are limited to our experiences, we understand that this research does not encompass every situation. However, we hope that, by sharing our own experiences, we can open a dialogue for researchers who have similar and different perspectives.

### 3. Researchers' Positionalities

To begin, we will share our distinct positionalities that influence our research. Discussing our positionalities is critical for two key reasons. First, the interactional nature of language documentation and reclamation, including the ethnographic methods we employed for our projects, means that our presence in our respective communities influenced participants and the data in a way that was sometimes difficult to predict. Secondly, our understanding lies with an idea that no one can be completely objective (e.g., Feyerabend 1993). As researchers who employ ethnographic methods on language in communities (Copland and Creese 2015; Creese 2010), we must be transparent about our backgrounds, which may include our epistemologies, values, and experiences, as well as how these may influence our interpretation of the data.

#### 3.1. Martha's Story

I am a biracial Japanese American who grew up in a close-knit multigenerational family, which included bilingual paternal grandparents who spoke Japanese and English. The intergenerational link was broken between my *nisei* ("second-generation") grandparents and *sansei* ("third generation") father, who did not acquire Japanese. My grandparents, great grandparents, great aunts and great uncles were incarcerated in the U.S. during World War II in internment camps. "Internment camp" as a term is sometimes contested, but I use it here as I feel that "war relocation camp" does not correctly describe the position of Japanese and Japanese Americans who endured the forced removal from their homes and the conditions at the camps where they resided for four years. My grandmother's family was sent to Jerome War Relocation Center in Arkansas, and my grandfather's family was sent to the Poston War Relocation Center in Arizona. Living in camps during WWII resulted in long-lasting generational trauma within the Japanese American community, including within my own family. During the war, my family's allegiance and loyalty were questioned, and they were treated as "the enemy" in the only home my grandparents had ever had. As a result, following their incarceration, and during their adolescence and early adulthood, many Japanese Americans felt that they had to work harder to prove themselves and their American-ness. Their efforts to assimilate fully into American culture included choosing to forgo transmission of Japanese language or even choosing not to give their children (i.e., my father's generation) Japanese names.

I consider myself and my siblings to be heritage speakers of Japanese, as we only acquired passive knowledge as children in the family domain and were raised as monolingual English speakers. By the time I was born in 1990, public American opinion had shifted, and being Japanese American was something to take pride in within our Japanese American community. I was raised to take pride in my Japanese heritage, but the intergenerational transmission link for the language had already been broken with my father's generation.

At university as a linguistics major, I took formal Japanese language courses and then moved to Tokushima Prefecture on Shikoku Island in Japan following my undergraduate graduation. I lived in Tokushima for three years (2012–2015), where I improved my Japanese language skills. While doing my PhD at SOAS, I used Japanese as a contact language when working with speakers of Setouchi Amami. I chose SOAS for its commitment to documentation and description of minority and under-documented languages and for its expertise in Asian languages. I was also drawn by the Endangered Languages Archive, which preserves over 500 languages and at the time was housed at SOAS. Few or no teaching materials are currently available to learn Setouchi Amami Ōshima, but I learned some in the field and continue to study on my own using my data.

Living in Japan from 2012 to 2015 granted me two important realizations that eventually led me to carry out language documentation in the Ryukyus. Firstly, I became aware of and was confronted with the prejudice against speakers of local varieties in Japan. I was told in Japan not to use words and phrases I had learned at home from my grandparents, who are descendants of farmers in the rural Japanese countryside of Wakayama Prefecture. I was told this is because the varieties spoken in the rural areas of Japan are considered to be "not sophisticated" or even the language of "country bumpkins". I also realized how much of an outsider I was in Japanese society, although I had been raised to feel deeply connected and proud of my Japanese roots. Once I was actually living in Japan and participating in Japanese society, I realized that my cultural thinking was at odds with my Japanese colleagues' expectations. Additionally, as a person with a non-Japanese parent, I often faced exclusion or exotification for my appearance. People with one ethnically Japanese parent and one non-Japanese parent are referred to as *hāfu* (from the English word for "half") in Japan. Despite growing numbers in the Japanese population, Japanese people with mixed heritage continue to experience both prejudice (such as bullying in school) as well as exotification and objectification (this is particularly true for women; Murphy-Shigematsu 2000).

The second thing that sparked my research interest was my own family's language shift. My own missed opportunity to acquire Japanese made me interested in language shift and language loss. Furthermore, the discrimination I experienced firsthand and witnessed (from the outside) being thrust upon other minorities in Japan brought my attention to minorities living within Japan, including Ryukyuans. Additionally, living in Japan provided the opportunity to visit the Ryukyus and meet people living there.

Zlazli (2019) asserts that Ryukyuan communities tend to perceive a boundary between insiders and outsiders, although community members generally do not show discrimination directly towards those they deem "outsiders" (p. 24). However, although I am American and always presented myself as such during my fieldwork, I did not detect that people ever felt uncomfortable with my presence/research or that I was unwelcome in the community. This interpretation is of course limited to my own singular perception of my interactions with community members.

Finally, some Ryukyuan people, particularly those who do not speak English or have much experience with foreigners, tend to have internalized self-orientalism (in the sense of Fishman and Gàrcia 2010). This has been observed in other parts of the Ryukyus as well, particularly by Hammine (2021), who defines self-orientalism as "a phenomenon in which individuals subconsciously devalue their own language in comparison to the dominant language here, Japanese" (p. 85). Therefore, there may have been a possibility of a power dynamic between the participants and me in this sense of self-orientalism, particularly since I am a native English speaker, as English is considered a prestigious language (see more on Martha's positionality in Tsutsui-Billins 2021).

### 3.2. Madoka's Story

I now consider myself to be an Indigenous scholar, as well as a learner and a new speaker of Ryukyuan linguistic varieties, Okinawan and Yaeyama. I was born and grew up in the Ryukyus. Before leaving the Ryukyus to be an exchange student in a U.S. high

school and then leaving for college in Tokyo, I had assimilated my identity as Japanese. Being originally from the Ryukyus, but educated at graduate schools outside the Ryukyus, I have experienced the ideology of homogenous Japan both within and outside the nation (Befu 2001).

Growing up in the Ryukyus in the 1990s, my family has always spoken to me in Japanese, and I went through my compulsory education in Japanese. Both my grandmothers and late grandfathers speak/spoke varieties of the Yaeyaman language and the Okinawan language as native speakers. However, they did not speak to my parents in these vernaculars. Due to prevailing homogenizing ideologies of Japan and a common colonial view to refer to Ryukyuan varieties as *hōgen* ("dialect") in the Ryukyus (Heinrich 2012; Lee 2009; Yasuda 1999), I realized only when I started studying at university in Tokyo that my grandparents were speaking Ryukyuan varieties. I became consciously Indigenous through years of learning and self-reflection, but I understand that the discourse of Indigeneity in Ryukyuan contexts needs further examination (e.g., Ginoza 2015; Kina 2013).

Having my roots in the Ryukyus, people from mainland Japan viewed me as *Okinawan*, asking me sometimes to speak in the "dialect" of Okinawa. Ryukyuan islanders tend to have different terms of expressing and identifying them from others, such as *Uchinaanchu* for people from Okinawa Island and *Yaimapïtu* for people from Yaeyama. Traditionally, islanders have been culturally, linguistically, historically, and spiritually different from "mainland" Japanese (see also Nakagawa 2020). People from the Ryukyus have been treated as a racialized minority within Japan. As an example, *jinruikan* ("the Human Pavilion") displayed two Okinawans with other racialized minorities in Japan as a part of 5th Domestic Industrial Exposition in Osaka in 1903. Historically, in Japan, people who were not considered part of the group of "dominant Japanese" include a historical underclass (the Burakumin), citizens of the empire (e.g., Koreans, Taiwanese, or Chinese), or Indigenous peoples of new Japanese territory (the Ainu and Ryukyuans). The Ainu in the north and Ryukyuans in the south of Japan were under government policy to be assimilated (*dōka*) into cultural invisibility (Weiner 2009; Oguma 1998). The cultural invisibility of Ryukyuans owes its roots to the multilayered history of the archipelago. The independent Ryukyu Kingdom only officially became part of Japan during the Meiji Period, when Japan was defining its borders. Since then, the Ryukyus have endured repeated mistreatment from Japan. In Amami Oshima, for instance, the Satsuma domain from Japan also created hardship for Amamians by levying heavy taxes and instigating the *sotetsu jigoku* ("cycad hell"), a famine induced by the Satsuma clan's taxation during the Edo Period and additional famine during the Depression after WWI, which led islanders to resort to a traditional, locally known process for removing the poison from cycad stems and seeds in order to use them as an emergency food (Maeda 2014, p. 237). The mistreatment from the nation also includes Japanese approval of American base placement on the islands (Okinawa prefecture has more bases than any other prefecture in Japan), and Japan's WWII use of Okinawa during the Battle of Okinawa, the only Japanese battle fought on land during WWII. During the battle of Okinawa, between 30,000 and 100,000 Okinawan civilians were killed, and many Okinawans were killed by Japanese forces due to fears of spying when Okinawans were speaking their local languages.

Nakagawa (2020), who writes as an Indigenous scholar from Tokunoshima in the Amami Archipelago, asserts that when islanders from Tokunoshima leave the island, they are often mistaken for Hawaiians, Filipinos or Samoans (p. 182). I have experienced being seen as the "Other" in both mainland Japan and abroad, similar to Nakagawa (2020), an Insider linguist from Tokunoshima. I am often mistaken for a foreigner or *hāfu* in mainland Japan, and often I am not seen as "Japanese" enough when I am abroad.

By encountering people who see me as the "Other" in mainland Japan and outside Japan, I realized that I did not know any of the vernacular languages of my families and my ancestors. I became a learner of these languages in my mid-20s more consciously through my experiences of being away from the Ryukyus. I started learning the vernacular languages more intentionally during my graduate study in Scotland where I also encountered

speakers of Scottish Gaelic in the northern part of Scotland. Through encounters with learners of Indigenous minoritized languages such as Scots, Gaelic and then Sámi languages, I became more conscious about where I am from. I learned that these Indigenous languages are experienced sometimes negatively by their speakers and learners. For my doctorate study, I chose to conduct research in Yaeyama language communities. My experiences with Sámi communities and Indigenous researchers I met in Lapland influenced my work, and their input was also crucial to conduct my doctoral research, centering Indigenous ways of seeing. This was a choice and involved a process of struggles and healing. By struggles, I refer to occasional moments of being used as a subject of research by linguistic researchers or being viewed as an "exotic" essentialized islander by the majority when presenting my work. By healing, I refer to moments such as being connected with my ancestors, family, and our local histories and traditions as well as creating a space/material/art with other learners of Ryukyuan. Since these varieties relate to my heritage, the process of research has been transformative in both positive and negative ways. Such struggles and healing to decolonize my heritage languages and cultures as a Ryukyuan speaker have shaped my earlier research (e.g., Hammine 2020). In this article, I attempt to localize the view of language endangerment by using a larger framework of decolonization via collaborative and relational methods to instigate a conversation in our field.

## 4. Why Collaboration Is Essential

Based on our different positionalities with different communities in the Ryukyus, we argue that collaboration is the best for both communities and researchers and that positionality–consciousness is not a dichotomy but rather a continuum. Firstly, collaboration welcomes community members to join researchers in the research "driver seats" and gives communities rightful control and proper ownership of their data, which are their own cultural heritages. Researchers and communities working together means that the communities' needs and agendas are at the forefront of research projects, rather than a side note to the researchers' agendas. Collaboration also makes the work more equitable, ethical and responsible because everyone's voices are being heard. To that end, collaborative research also enables a "trauma-informed approach" (Kono and Switzler 2021) to language documentation and revitalization. Using their own research with Indigenous language documentation and revitalization in North America, Kono and Switzler (2021) show that community members (collaborators) can learn better when researchers have awareness of collaborators' past and present experiences and the effects of those experiences on their well-being and their ability to engage with the learning materials. By sharing and understanding the experiences of the community, they ask for inclusion of healing/trauma-informed approach to language documentation and revitalization (Kono and Switzler 2021). The healing-informed approach, originally adopted from the fields of nursing and psychology (e.g., Selwyn and Lathan 2020; Ward 2020), is guided by four assumptions, known as the "Four Rs"—*realization* about trauma and how it can affect people and groups, *recognizing* the signs of trauma, having a system which can *respond* to trauma, and *resisting* re-traumatization (Kono and Switzler 2021). This means that researchers need to be intentional in promoting a safe environment that cultivates connectedness, empowerment, and recovery.

Furthermore, representation in research is essential—the more local people are involved in language research, the more open the field will be to others. If local people see others like them successfully participating as researchers, this can increase the numbers of Insider researchers through empowerment. Communities will be more invested in the research if it is being driven by the community's interest or the community's agenda, and work is more "rewarding" and has more longevity—that is, the research can continue for many years beyond the time that the scholar is working in the community. Furthermore, community members can gain skills from linguistic training that may be transferable to their chosen career paths. For example, language documentation involves using computers, and learning how to use computer software may be advantageous for community members.

For language documentation, computer programs such as ELAN, FLEx, and Praat—which might be unfamiliar at first to users—may pose an opportunity for community members to advance their computer literacy skills. Even using more basic computer programs, such as Microsoft Word and Microsoft Excel, can be useful. In some cases, these more basic programs might be even more applicable for community members, as they are ubiquitous and present in many fields and industries beyond linguistics.

A further advantage of collaborative research is that horizontal learning can take place by deconstructing a binary of fixed categories. Rather than the researcher being regarded as the "expert" and the community members as the "collaborators", taking a collaborative approach means recognizing the expertise that community members bring to the table. Community members will have cultural and linguistic knowledge from the first day of the project that a new outsider researcher will not (and may never) have, so working together allows research to be more nuanced and in-depth with the help of insiders. This way, community members can teach the linguist and the linguist teaches speakers. Furthermore, working together means the research/project benefits from more perspectives and contacts, the work is more well-rounded, and the resulting corpus is richer.

Collaboration is a way to relinquish control and let others decide. Collaboration, thus, often slows things down in some ways—this means extra planning must be included in the project because it is not just one person deciding or being depended on to make grant and university deadlines. Others are also involved, and this means other people's commitments (families, jobs, etc.) should also be considered.

## 5. Considerations Specific to the Ryukyuan Context

Collaboration is not easy in the Ryukyuan context since different individuals in different language communities have different political agendas and identities. In the case of Madoka, who is partially also an insider as a heritage speaker of Yaeyaman and Okinawan, carrying out language documentation and revitalization work could be seen as political. Due to the ambiguous positions of Ryukyuan, often viewed as a dialect (*hōgen*) by native speakers in communities, I, (Madoka), as a researcher, have to be careful not to overlook political issues. As a (partial) Insider researcher, I must listen to community members who are living in Yaeyama. In my father's home village, the research process went relatively smoothly. In other communities in Yaeyama, complex power relationships between villages and my position as someone who has higher education from Europe made it difficult and caused some resistance from community members. Within Yaeyama, for instance, the Shika-aza linguistic variety is sometimes seen as better than the Miyara variety. As a speaker of Miyaran, I sometimes feel resistance from speakers of Shika-aza. Histories of conflicts between different villages also make it difficult to carry out my work. A political agenda is always at stake.

We also identify the multiple layers of colonial history in the Ryukyus as an important issue to understand. When working with language users, political agendas of each community's members sometimes conflict and make research difficult. Although the last decade saw a significant change in the identification of Ryukyuan as language in its own right, islanders in the Ryukyus in general tend to refer to these languages as *hōgen* or dialects of Japanese. For instance, Heinrich et al. (2009) write that the situation in the Ryukyus is complex:

> The US occupation of Uchinaa after World War II, which—at least formally— ended in 1972, marks the final stage in the fading of the Luchuan languages. In their attempts to separate Uchinaa from mainland Japan, Americans emphasized the distinctiveness of the Luchuan languages and cultures and encouraged their development. This US policy of dividing Luchuan from Japan, however, backfired and gave rise to a Luchuan Japanization movement. Today, even the remaining— mainly elderly—Luchuan language speakers generally refer to their languages as hōgen, i.e., Japanese 'dialects', accepting in so doing the downgrading of their heritage languages for the assumed sake of national unity. (p. 2)

Previously, Nakayama and Ono (2013) showed that the collaborative model is not "easily applied" to all contexts. Working on the Ikema dialect of the Miyako language, Nakayama and Ono (2013) recognize these difficulties in community collaboration in the Ryukyus, mainly that speakers do not always have a "a clear sense of or are not particularly interested in examining its current state, nor do they know what they want to do for their language". In my (Martha's) case, I have also found this to be true of some community members. I have also encountered differing opinions from community leaders regarding what the best way forward is. Should someone offer Amami language classes focusing on conversation? Should the responsibility fall on younger people to take an interest and seek out traditional Amami arts to learn the language? Is there any hope for local languages? I have seen these questions met with differing responses in the Setouchi community. Anecdotally, I have also heard of similar discussions and disagreements in other Ryukyuan communities. This issue relates to Nakayama and Ono's (2013) point that individuals in communities are not homogenous of thought—thus "it is rather difficult to identify the needs of the 'community' as a whole". Some community members feel distinct from mainland Japanese society while others do not.

In the Ryukyus, historical assimilation of Ryukyuan speech practices to Standard Japanese has continually subordinated Indigenous groups. When researchers are not interested in carefully examining "their" experiences in the Ryukyus, it becomes hard to judge the needs of the "community". The difficulty of establishing language reclamation might be related to the lack of understanding in Japanese society about ethnicities in Japan. In Japanese, islanders of the Ryukyus are *Okinawa-jin* ("Okinawan"), *Nihon-jin* ("Japanese") or *Ryukyu-jin* ("Ryukyuan")—regardless, I (Madoka) do not know the correctly accepted way of identifying myself. As Nakayama and Ono (2013) assert, there seems to be a lack of desire to establish the traditional languages as separate languages from Japanese among speakers and community linguists. With the effect of Japanese colonialism, where Japanese language was used as a means to unify or absorb the Ryukyus into Japan through standardized monolingual education in Standard Japanese from the Meiji Period (Osumi 2001; Kondo 2008), some community members might have lost social, cultural, and political opportunities to consider Ryukyuan languages as separate from Japanese repeatedly and intergenerationally.

While the traditional "endangerment" approach in language documentation and revitalization may entail colonial attitudes toward languages and people, the collaborative research approach instead prioritizes the relationships with the communities, people, speakers, the Earth and beyond as Indigenous Research Methodologies emphasize relationality (e.g., Chilisa 2019; Smith 2012; Tsikewa 2021; Wilson 2008). In Indigenous worldviews, relationships with nature and the Earth are seen as important. Ryukyuan worldviews that are embedded in Ryukyuan languages and cultures also have Indigenous epistemologies that people have based their knowledge on (see Guay in this volume). As one example, during the COVID-19 crisis, local islanders started a practice of singing *Tinsagu nu Hana*[1] ("The Balsam Flower," a Northern Okinawan song) while washing their hands (Hantagawa Kominkan 2020). By using Indigenous songs sung predominantly in the Okinawan language, local knowledge is practiced. For instance, in this song, *chimu* (literally, "liver" but spirit, mind, and soul in the wider sense) refers to a concept in the Ryukyus to consider relationships with others, communities, nature and land. This concept, *chimu*, also entails trust, respect and harmony beyond relationships. By recognizing *chimu* as an important concept when carrying out research, researchers work consciously based on a holistic understanding of language, their speakers, leaners, language communities, land, and the Earth. Eventually, this practice benefits science by taking a relational approach instead of a (colonial) individualistic approach to Ryukyuan linguistics.

## 6. Our Stories and Recommendation for Future Research

This section will cover our recommendations on insider/outsider collaboration, starting with the first step of listening to people. By compassionately listening (Hammine

2022; Rehling 2008; Furman 2009), people on all parts of the insider/outsider spectrum can find collaborators. Along with listening, we recommend making yourself known so that interested people will come to you. We should be trying to work with volunteers, not recruits. Do not leave things until you are in a desperate situation with tight deadlines and have grant agencies with big expectations; instead, it is much better to start small and slowly. As researchers, we can go to the community with humility first and find out if they are even interested in working with us, rather than assuming that people will be interested in language research. If it is impossible to physically go to the community island, one can make contact online through a "friend of a friend" (Milroy and Milroy 1978). One fortunate fact about the Ryukyus is that the technological communicative issues that other areas face are not an issue in the Ryukyus where Wi-Fi and smart phones are ubiquitous amongst members of every generation.

Once contact has been made and interested parties have come together, one can collaboratively plan the research with the question "what aims suit all parties?" in mind. This utilizes Indigenous methodology by moving "beyond an 'Indigenous perspective in research' to 'researching from an Indigenous paradigm'" (Wilson 2001, p. 175) and drawing on "relational accountability" (Wilson 2008, p. 40) to include or focus on revitalization aspects.

*6.1. Martha's Experience*

My experience with collaboration in the Ryukyus involved doing language documentation training with community members. During my PhD, I had a broad topic (honorifics) that I hoped to document and research in Setouchi, Amami. Once I arrived in Amami, I listened to community members' input on my topic and took in their recommendations. Many speakers told me about verbal honorifics in Setouchi Amami, and this influenced my decision to research verbal predicates specifically for my PhD thesis. These verbal predicates are well-known in Setouchi Amami and often appear in formulaic expressions, so by focusing on this aspect of the language, the project could document something that matters to the community and is present in the current speech community, including less fluent and younger speakers.

During my first visit to Setouchi, I collected conversational data between different generations of speakers. One setting where I wanted to collect data was at retirement homes because this was a place where speakers of differing ages and fluency communicated on a daily basis. Younger Setouchi Amami speakers in their 50s and early 60s worked at these retirement homes as staff members, and much older fluent Setouchi Amami speakers lived at the retirement homes as residents. Unfortunately, the first time I visited Amami was during the winter months and due to precautions to protect against influenza, extra visitors were not allowed into the retirement homes. As a non-family and non-staff member, I was not allowed to meet elders in the retirement home. Thus, this situation became an excellent opportunity to train community members in language documentation data collection. Retirement home staff received basic recording, equipment, and consent training from me and then recorded themselves interacting with their colleagues and the elder residents. Not only was it satisfying for myself and the staff members to transfer and learn a new skill, it also reduced the obstacle of the observer's paradox (Labov 1991) because speakers were recording themselves going about their daily lives without an outsider standing over them and recording them.

One crucial element to this approach was using non-obtrusive equipment. A lapel microphone in addition to a video camera would be ideal to collect all manners of communication, including non-verbal cues (such as gestures like bowing). However, extra equipment would be cumbersome, and I found that participants were more at ease with using either a single Zoom H4n audio recorder or a recording watch which could be activated with a single button. Everyone in the community was notified that recordings would be taking place, and participants were asked again after the recordings were completed to confirm they were happy with the recordings that had been collected to ensure informed consent.

*6.2. Madoka's Experience*

During my PhD, I started collaborating with different researchers who identify themselves as "outsider researchers". By outsider researchers, I refer to those who do not have their heritage in the Ryukyus. At the beginning, it was a challenge to work with outsider linguists because their understanding of languages was different from mine. Collaboration with outsider researchers was possible and made me realize the importance of multiple perspectives to compare the situation of the Ryukyus with others as a researcher. Before starting any collaboration, I felt that I did not have any resources or anyone to consult with on my heritage languages. By collaborating with linguists who have expertise in languages, I learned how to carry out documentation and descriptions of the Ryukyuan languages. I also participated in different conferences where descriptive linguists discussed their research. After learning from their work, I gained more insights about Ryukyuan.

I also visited different communities in Yaeyama with "outsider" researchers. During my fieldwork, I am sometimes asked if I come to peoples' houses alone or if I bring someone else. When I come to visit peoples' houses with outsider researchers, I have to pay attention to community members' reactions toward us. I have noticed that often people choose what to tell us, and people tell me different things when I am alone. I have also noticed that sometimes some of the so-called "semi-speakers", who are passive bilinguals in their heritage languages, tend to feel intimidated or ashamed of not being able to "help" researchers who want to focus on data of native fluent speakers. By coming to the Ryukyus as an outsider who is trained to document local languages, it is essential to pay attention to voices of community members who are not included on the basis of researchers' definitions of "native" and "non-native". Understanding researcher positionality is crucial to collaboration. Although it is difficult to raise these questions for outsider researchers, it is important to ask questions such as: Who are you? Who do you identify with? What do these languages mean to you? What do these languages mean to the community?

There is an issue around public awareness about gender differences of researchers. As a female researcher, I am often seen as a helper or an assistant of a male researcher when I go to visit peoples' houses in the Ryukyus. As an Indigenous Ryukyuan, I am supposed to follow community rules around the house—taking care of the work in the kitchen, serving food to guests, and so forth. I sometimes feel caught between expectations and being critical about patriarchy in communities. Based on my experiences, I encourage future collaboration in the Ryukyus to focus on unheard voices, including non-native speakers, female members, and individuals who are marginalized within conflicting binaries of Indigenous/non-Indigenous.

Ryukyuan community members tend to avoid conflicts with each other, so they may not feel safe sharing everything they feel with strangers, particularly someone who comes to the community but does not live there. Therefore, when working in the Ryukyus, it is extremely important to be able to listen to community members and people in the community. If researchers point out how "wrong" people speak languages or how "wrong" community members work in language revitalization projects, speakers and learners could be further silenced and re-traumatized. Compassionate listening practice, as used in other fields of psychology, could hence work well when applied by researchers in the Ryukyus (e.g., Furman 2009; Kimble and Bamford-Wade 2013; Rehling 2008). I have observed how the understanding of intergenerational trauma in communities and positionalities of researchers could transform effects of colonialism at deep levels (Hammine 2022). By reflecting positionalities and responsibilities, researchers can transform principles of research. Decolonization, in this sense, relates not only to community members but to everyone, including researchers.

## 7. Conclusions

In this article, we highlighted the shift from research *on* the Ryukyus based on West-centric modernist understanding of languages to research *with* the Ryukyus. Throughout the article, we discussed the importance of collaboration based on relationality, which

is also at the center of Indigenous Research Methodologies as described by many other scholars in other contexts around the world (Chilisa 2019; Smith 2012; Wilson 2008). A decolonizing agenda must support language communities to create and reclaim their own frameworks, languages, theories, epistemologies, philosophies, and methodologies. To shift perspectives of researchers to that of becoming community-centered, we argue that we must listen to people who have experienced language endangerment as their lived experiences (i.e., compassionate listening practice, trauma-informed approach). To do that, we ask fellow researchers to understand our/their own positionalities when researching *with* Ryukyuan communities—who you are, where you come from, where you are headed, how you relate to those you are doing research with.

For readers who wonder whether they should be attending more meaningfully to the work of institutionalizing Indigenous studies, collaboration might be one answer. With established networks and collaboration with experts, it is possible to have established networks and good, strong collaboration that give one a place to be and a sense of a shared community. Sharing, as a crucial concept of Indigenous epistemologies in other contexts (e.g., Porsanger et al. 2021; Kovach 2005; Wilson 2008), is at the center of the *relational* approach in research that we propose in the context of the Ryukyus. In this way, sharing data becomes natural and crucial for researchers rather than dividing languages from people in the communities that we work with. Hence, we can work *with* Indigenous communities with a sense of data sovereignty for Indigenous populations.

We wrote this article as a dialogue between the authors to start a broader conversation and interactions between researchers in Ryukyuan language research. This paper itself is also an example of insider–outsider collaboration because, as seen in our positionalities, both authors fall within the continuum of outsider/insider. Throughout the article, we emphasized the importance of a relational approach in research, including insider/outsider collaboration and language documentation based on community needs. We hope our publication will act as a guide and promotion for collaboration between researchers with different positionalities in the Ryukyuan contexts. Through a collaborative relational approach in research, we argue that future research should be *with/for* the Ryukyus with a purpose to emancipate language communities, rather than research *on* the Ryukyus. Ultimately, we argue that decolonization is not only for Indigenous peoples, but also for everyone, including researchers who conduct research with Indigenous language communities

**Author Contributions:** Conceptualization M.H. and M.T.B.; formal analysis M.H. and M.T.B.; writing—original draft preparation M.T.B. and M.H.; writing—reviewing and editing M.T.B. and M.H. All authors have read and agreed to the published version of the manuscript.

**Funding:** This research was funded (partially) by Foundation of Endangered Languages (FEL).

**Institutional Review Board Statement:** The study was conducted in accordance with the declaration of Helsinki and approved by the Institutional Review Board of our respective graduate schools (University of Lapland and SOAS University of London).

**Informed Consent Statement:** Informed consent was obtained from all subjects involved in the study during both authors fieldwork.

**Data Availability Statement:** The data presented in this study are available on request from the corresponding author. The data are not publicly available due to privacy.

**Conflicts of Interest:** The authors declare no conflict of interest.

## Notes

[1] My mother (Author 1) shared this song with me one day during the COVID-19 crisis. After hearing about this practice from her, I noticed many local newspapers and community centers emphasizing the importance of traditional Okinawan songs sung when people wash their hands. This is an example of how Indigenous knowledge through songs is practiced.

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
