# Peer review of "Collaborative Ryukyuan Language Documentation and Reclamation"

_languages, doi:10.3390/languages7030192_

Round 1
Reviewer 1 Report
Please see the attached file.

Author Response
Thank you very much for your comments.
We immediately understood your two main points for our paper.
Regarding your point 1, we tried to explain our intention and what we mean by "endangered" approach and "endangered" paradigm in our secound paragraph. In the same paragraph, we also explained how the traditional research have dealt with endangered languages in Japan and described how we aim for changing this approach by using Indigenous decolonial appraoch, focusing of collaboration and relationality.
Related to you comment 2, we also included one sentences to explain the dominant "endangerment" appraoch successfuly raised peoples' awareness, however, it does not successfully accepted by communities since it takes an approach which was imposed on the langauge communities. With this article, we aim to start a dialogue with different researchers both domestically and globally, community members, langauge activists to improve research approach in langauge endangerment.
We hope that our changes would make our intention clear and make our arguments understandable without confusing readers.
Thank you very much for your comments. We appreciate all the comments you provided us.
Sincerely,
Authors
Reviewer 2 Report
Please see attached file

Author Response
Thank you so much for reviewing our article and providing us with constructive comments on our paper.
Based on your review, we revised our paper. Firstly, and most importantly, after reading your comment that our article sounded like collaboration was new to language documentation, we added sentences in paragraph 2 in the introduction saying that we are aware of collaborative methods are becoming more and more "standard" in other endangered language communities. We also included new references, such as Leonard 2018 and Leonard 2019 in our revision. We agree with you completely that collaboration and centering Indigenous knowledge system have been already pointed out by other scholars who work in other communities. We hope this is clear in our article now.
In addition, we also changed small language problems that you mentioned in your review. Thank you for reading our article so carefully. One point that you mentioned about the term, hogen, we still used the same word despite your suggestion to use "dialect" instead. This is because we are not sure if the term "dialect" is the exact translation of hogen. In Japanese, there is another word "ben" which could be translated into dialect in English. We went through all the parts where we had the term, and in some places we still used hogen while in other parts, we used "dialect" as you suggested. Thank you so much.
All of your comments are very much appreciated and we would like to thank you again for such a careful and constructive comments on our article.
Sincerely,
Authors
Round 2
Reviewer 2 Report
See attached file

Author Response
Thank you for your comments. We have had our paper professionally proof-read and corrected using the "tracked changes" feature in Microsoft Word.